# Unveiling the Future of Cardiac Care: A Review of Gene Therapy in Cardiomyopathies

**DOI:** 10.3390/ijms252313147

**Published:** 2024-12-06

**Authors:** Damiano Venturiello, Pier Giorgio Tiberi, Francesco Perulli, Giulia Nardoianni, Leonardo Guida, Carlo Barsali, Carlo Terrone, Alessandro Cianca, Camilla Lustri, Matteo Sclafani, Giacomo Tini, Emanuele Barbato, Beatrice Musumeci

**Affiliations:** 1Cardiology, Clinical and Molecular Medicine Department, Sapienza University of Rome, 00189 Rome, Italy; damiano.venturiello@uniroma1.it (D.V.); piergiorgiotiberi@yahoo.com (P.G.T.); francescomaria.perulli@gmail.com (F.P.); giulia.nardoianni@uniroma1.it (G.N.); leonardo.guida@uniroma1.it (L.G.); carlo.barsali@uniroma1.it (C.B.); carlo.terr@gmail.com (C.T.); cianca.alessandro1995@gmail.com (A.C.); cami.lustri@gmail.com (C.L.); matteo.sclafani18@gmail.com (M.S.); giacomo.comotini@gmail.com (G.T.); emanuele.barbato@uniroma1.it (E.B.); 2Royal Brompton and Harefield Hospitals, Guy’s and St Thomas’ NHS Foundation Trust, London SW3 6PY, UK

**Keywords:** cardiomyopathies, gene therapy, CRISPR/Cas9, adeno-associated virus (AAV), transgene expression, clinical trials in cardiomyopathies, vector delivery, gene editing, gene replacement

## Abstract

For years, the treatment of many cardiomyopathies has been solely focused on symptom management. However, cardiomyopathies have a genetic substrate, and directing therapy towards the pathophysiology rather than the epiphenomenon of the disease may be a winning strategy. Gene therapy involves the insertion of genes or the modification of existing ones and their regulatory elements through strategies like gene replacement and gene editing. Recently, gene therapy for cardiac amyloidosis and Duchenne muscular dystrophy has received approval, and important clinical trials are currently evaluating gene therapy methods for rare heart diseases like Friedreich’s Ataxia, Danon disease, Fabry disease, and Pompe Disease. Furthermore, favorable results have been noted in animal studies receiving gene therapy for hypertrophic, dilated, and arrhythmogenic cardiomyopathy. This review discusses gene therapy methods, ongoing clinical trials, and future goals in this area.

## 1. Introduction

Cardiomyopathies represent a miscellaneous group of heart diseases characterized by structural and functional abnormalities of the heart muscle, often leading to heart failure, arrhythmias, and sudden cardiac death [1]. Traditional treatment modalities, including medications, lifestyle modifications, and, in some cases, surgical interventions, have offered varying degrees of success but fall short in addressing the underlying genetic causes of these disorders. As our understanding of the genetic basis of cardiomyopathies increases, gene therapy emerges as a revolutionary approach with the potential to transform the landscape of cardiac care [2].

Gene therapy, the technique of introducing, removing, or altering genetic material within a patient’s cells to treat or prevent disease, has shown remarkable promise in other fields of medicine. Since the first approved human gene therapy trial occurred in 1989 utilizing a retroviral vector to insert a neomycin resistance gene for treating metastatic melanoma, scientists have been working to develop safe, durable, and efficient methods to modify and control human cell genome for therapeutic purposes in various hereditary conditions [3]. In the context of cardiomyopathies, this innovative approach aims to correct or compensate for the genetic mutations that lead to these debilitating conditions. By targeting the cause of the disease at the molecular level, gene therapy offers a more personalized and potentially curative alternative to conventional treatments.

This review aims to analyze the current state of gene therapy for cardiomyopathies, examining recent advancements, ongoing clinical trials, and the challenges that remain.

## 2. Materials and Methods

### 2.1. Literature Search

The search of the literature on the use of gene therapy in cardiomyopathies has been fundamental for the development of this review. The research was mainly conducted on online scientific platforms and journals of cardiology and molecular medicine. The keywords searched were: Cardiomyopathies, Gene therapy, Adeno-associated virus (AAV), Clinical trials In Cardiomyopathies, CRISPR/Cas9, Transgene expression, Vector delivery, Gene Editing and Gene Replacement.

### 2.2. Inclusion and Exclusion Criteria

Inclusion criteria included scientific articles that discussed the use of gene therapy, in general, and more specifically in cardiomyopathies. Both preclinical and clinical studies were included. Articles that did not specifically address gene therapy and cardiomyopathies were excluded.

### 2.3. Analysis Process

Once the articles of interest were identified, a thorough analysis was conducted to determine which clinical trials were the most intriguing on the subject, also from a therapeutic perspective. The main trials that have marked the history of gene therapy in cardiomyopathies up to 2024 were described.

## 3. Results

### 3.1. Types of Gene Therapy

Cardiomyopathies can result from loss-of-function (LOF) and gain-of-function (GOF) mutations in genes crucial for cardiac function [4]. Nonsense, frameshift, and sometimes missense mutations are LOF mutations that can result in either no protein being produced or in a nonfunctional protein being produced (Figure 1A). On the other hand, missense variants frequently result in GOF mutations, which produce proteins with changed or increased activity compared to their wild-type counterparts (Figure 1B).

These genetic changes may be the target of efficient gene therapy techniques. Based on their intended treatments, these techniques can be broadly divided into gene replacement and genome editing.

To alleviate the impact of a LOF mutation, gene replacement therapy seeks to substitute a malfunctioning gene with a healthy version [5]. This can be accomplished by inserting a functioning version of the gene into the cell through the use of plasmid DNA or through viral vectors [6,7]. Comparing plasmid-based therapy to viral vector-based therapy, the former reduces carcinogenic hazards while increasing the possibility of long-term effects [8]. On the other hand, transporting plasmids to the nucleus might be challenging [9,10]. Several types of viral vectors derived from both DNA and RNA viruses have been developed for use in gene therapy. The currently used viral vectors in gene therapy in cardiomyopathies are adeno-associated viruses (AAV), which are small, non-enveloped, single-stranded DNA (ssDNA) viruses that can contain 4 kb of foreign DNA [11]. While AAV does not induce toxic or harmful reactions, repeated injections of AAV vectors have led to strong immune responses, which diminish the efficiency of delivery and expression of the transgene [12]. This scenario could be solved by pharmacological modulation of the immune response. In particular, adding 28 S ribosomal DNA homology sequences to AAV vectors increased AAV integration frequency by 30 times [13]. Researchers discovered through various animal models that serotype 9 (AAV9) is the most effective for transferring genes to the heart [14].

Recently, the development of MyoAAVs and the identification of myotropic AAV through engineered AAV capsids also show an enhanced and more selective therapeutic effect on skeletal and heart muscles [15].

Nonviral vectors have been developed as alternatives to viral vectors to address their disadvantages. These nonviral options include materials such as polymers, lipids, inorganic particles, and hybrid strategies. Many of these systems have a positive charge and can pair with negatively charged DNA to form complexes. These complexes have the ability to bind to negatively charged components of cell membranes, facilitating internalization [16].

Genome editing, as opposed to gene replacement therapy, can correct both gain- and loss-of-function mutations by altering the cell’s natural genomic sequence [17]. Precise DNA base pair alterations, including conversion, deletion, insertion, and combinations of these, are possible because of CRISPR/Cas9 technology. CRISPR/Cas9 is made up of a guiding RNA (CRISPR) that aligns with the specific target gene, along with Cas9, an enzyme that creates double-stranded breaks in DNA (DSBs). The repair of these DSBs can occur via nonhomologous end-joining, which results in frameshift mutations that interfere with protein synthesis, or through homology-directed repair (HDR), which precisely fixes the damaged DNA but is only effective in cells that are dividing [18].

There are new types of gene therapy techniques that modulate gene expression through gene silencing. This is possible through small interfering RNA (siRNA) and Antisense Oligonucleotides (ASO), which determine exon skipping.

Genes are composed of exons, segments that need to fit together with their adjacent pieces like a puzzle [19]. When a gene is transcribed, the exons are combined in a process known as splicing. If an exon is removed, the components are unable to connect. The exon-skipping method employs small molecules called antisense oligonucleotides to assist cells in bypassing a particular exon during splicing. This enables cells to link a different combination of exons, resulting in a protein that is shorter than normal but potentially still functional [20].

Furthermore, siRNA is a type of double-stranded RNA that usually measures 20 to 24 base pairs, much like micro RNA, and functions in the RNA interference mechanism. It disrupts the expression of particular genes that have matching nucleotide sequences by breaking down mRNA following transcription, thereby inhibiting translation [21].

### 3.2. Gene Therapy in Cardiomyopathies

#### 3.2.1. Hypertrophic Cardiomyopathy

Hypertrophic cardiomyopathy (HCM) is a fundamental heart muscle condition that clinically affects around 1 in every 500 people. Its primary characteristic is an unexplained thickening of the left ventricle, which cannot be linked to abnormal pressure or volume loads. This disorder is mainly passed down through families as an autosomal dominant trait, often exhibiting incomplete penetrance [22].

The eight most prevalent genes that code for the proteins in both thick and thin filament sarcomeres include: beta-myosin heavy chain (*MYH7*), alpha-tropomyosin (*TPM1*), cardiac troponin T (*TNNT2*), cardiac myosin binding protein C (*MYBPC3*), regulatory myosin light chain (*MYL2*), essential myosin light chain (*MYL3*), cardiac troponin I (*TNNI3*), and cardiac alpha-actin (*ACTC1*). Among these, two key genes—*MYH7* and another—together represent nearly 75% of all known mutations [23].

The application of gene replacement in HCM is promising, especially concerning *MYBCP3* mutations, as these typically result in low levels or a complete lack of mutant proteins, leading to haploinsufficiency [24]. Mearini and colleagues demonstrated that administering wild-type *MYBCP3* cDNA to homozygous *MYBCP3*-targeted knock-in mice inhibited cardiac hypertrophy and dysfunction while also elevating the levels of *MYBCP3* protein [25]. The delivery was carried out with the vector AAV9 that encodes *MYBCP3* in mice that were one day old. Over the 34-week observation period, there was an increase in both *MYBCP3* mRNA and *MYBCP3* protein levels. Another study showed that introducing the *MYBCP3* gene through AAV effectively restored the contraction strength in artificial heart tissue created from cardiac cells of newborn mice with targeted knockout of *MYBCP3* [26]. This idea was additionally validated in human-induced pluripotent stem cells obtained from a patient with HCM who had a truncating mutation in the *MYBCP3* gene [27]. The delivery of full-length *MYBCP3* cDNA via AAV into cardiomyocytes derived from patient-specific human-induced pluripotent stem cells led to increased levels of *MYBCP3* mRNA and *MYBPC3* protein seven days post-transduction [27]. A study conducted by Monteiro da Rocha and colleagues also found favorable outcomes when employing gene replacement techniques in cardiomyocytes derived from human embryonic stem cells with a truncating *MYBPC3* mutation [28]. In this research, the delivery of the *MYBPC3* gene via AAV vectors inhibited the onset of hypertrophy and disorganized myocardial structure while also enhancing calcium signal conduction in cardiomyocytes affected by HCM [28].

There is only one ongoing clinical trial regarding HCM: MyPEAK-1 (NCT05836259) (Table 1) [29]. This is the first non-randomized and open-label study involving humans, aimed at assessing the tolerability, safety, and pharmacodynamics of TN-201, a recombinant AAV9 containing the *MYBPC3* transgene, in adult individuals suffering from symptomatic non-obstructive HCM associated with *MYBPC3* mutations. The trial will recruit a minimum of 6 and up to 30 patients, who will be assigned to 2 escalating dose groups. Every participant will receive the active medication. The follow-up period for the patients will extend over 5 years after a single administration of TN-201. The primary outcome measures will include the frequency and severity of adverse events, as well as the incidence of serious adverse events linked to the study medication. The secondary outcome measure will assess the change from baseline to Week 52 in the Kansas City Cardiomyopathy Questionnaire Clinical Summary Score (KCCQ-CSS).

Direct genome editing by the CRISPR-Cas9 technology has been tested in HCM patients. In the research conducted by Ma et al. [30], the authors assessed a male patient with HCM who had a family history of the condition linked to a GAGT deletion in exon 16 of the *MYBPC3* gene. They reported that stimulating an inherent DNA repair mechanism specific to germline cells was able to rectify germline mutations. Nevertheless, these results are still subject to debate.

Exon-skipping strategies were similarly applied in HCM. Gedicke-Hornung et al. [31] assessed the potential of antisense oligonucleotide-mediated exon skipping as a gene therapy method in *MYBPC3*-targeted knock-in mice. These mice possessed a homozygous guanine (G) to adenine (A) transition at the final nucleotide of exon 6, leading to the production of defective mRNAs. By utilizing specifically designed antisense oligonucleotides and AAV9-mediated delivery to induce skipping of both exon 5 and exon 6 of *MYBPC3*, the study successfully reversed cardiac dysfunction and inhibited left ventricular hypertrophy in the mice. The advancement of allele-specific silencing through RNA interference presents an alternative approach to treating autosomal-dominant disorders by focusing on the mutated allele either in the laboratory or in living mice. In the case of HCM, this method has been employed to remove the mutant allele and slow down the development of cardiomyopathy in mice with a targeted *MYH6* knock-in [32].

#### 3.2.2. Cardiac Amyloidosis

Cardiac amyloidosis is characterized by the accumulation of improperly folded proteins in the extracellular matrix of the heart muscle. The majority of cases are associated with either monoclonal immunoglobulin light chain amyloidosis (AL) or transthyretin amyloidosis (ATTR), which can present in genetic (ATTRv) or non-genetic (ATTRwt) forms [23].

Recently, two gene silencing therapies have been authorized to reduce transthyretin (TTR) production in the liver by over 80%. These include siRNAs and ASOs [33].

Patisiran is a siRNA encapsulated in lipid nanoparticles, which is given i.v. every three weeks. After the intravenous infusion, patisiran siRNAs bind to apolipoprotein E, enter the hepatocytes, and degrade TTR mRNA in the cytoplasm. The APOLLO-A study (NCT01960348) was a phase 3, double-blind, placebo-controlled trial that evaluated patisiran administered at a dosage of 0.3 mg/kg every three weeks over a period of 18 months. This trial involved patients suffering from TTR-Mediated-Polyneuropathy [34]. The primary outcome measure was the change in the modified Neuropathy Impairment Score + 7 (mNIS + 7). At baseline, the average mNIS + 7 score was 80.9 ± 41.5 for those receiving patisiran, while the placebo group had an average score of 74.6 ± 37.0. After 18 months, the adjusted mean change from the baseline was −6.0 ± 1.7 for the patisiran group, contrasted with 28.0 ± 2.6 for the placebo group, leading to a significant difference of −34.0 points (*p* < 0.001). Patisiran also had a favorable effect on gait speed and modified Body Mass Index (BMI). At the end of the 18-month period, the adjusted mean change in gait speed was 0.08 ± 0.02 m/s for the patisiran group, compared to −0.24 ± 0.04 m/s for the placebo group, resulting in a difference of 0.31 m/s (*p* < 0.001). For modified BMI, the changes were −3.7 ± 9.6 for patisiran and −119.4 ± 14.5 for placebo, yielding a difference of 115.7 (*p* < 0.001). Approximately 20% of participants receiving patisiran and 10% of those on placebo experienced mild to moderate infusion-related reactions; however, the overall incidence and types of adverse events were similar between the two groups. Patisiran received FDA approval in 2018 and is indicated for the treatment of the polyneuropathy associated with ATTRv in adults.

The APOLLO-B trial (NCT03997383) was a phase 3, randomized, double-blind, placebo-controlled study in which participants with ATTRv or ATTRwt were allocated in a 1:1 ratio to receive either patisiran (0.3 mg per kilogram of body weight) or a placebo, administered every three weeks for a total of 12 months [35]. A total of 360 individuals were randomly assigned to either the patisiran group (181 patients) or the placebo group (179 patients). The primary endpoint was the change from baseline in the Six-Minute Walk Test (6-MWT) at month 12. At the end of the 12-month period, the decrease in distance walked during the 6-MWT was less significant in the patisiran group compared to the placebo group (Hodges–Lehmann estimate of the median difference: 14.69 m; 95% confidence interval [CI]: 0.69 to 28.69; *p* = 0.02). Additionally, scores on the Kansas City Cardiomyopathy Questionnaire Overall Summary (KCCQ-OS) improved for the patisiran group, while they worsened for the placebo group (least-squares mean difference of 3.7 points; 95% CI: 0.2 to 7.2; *p* = 0.04). No significant advantages were observed for the secondary endpoints, which included a composite assessment of all-cause mortality, cardiovascular events, and changes in the 6-MWT distance over the 12-month period analyzed by win ratio. This trial indicated that treatment with patisiran over a 12-month span helped preserve functional capacity in patients with ATTR cardiac amyloidosis.

Vutrisiran is a next-generation siRNA that demonstrated a maximum reduction of 80% in TTR levels over 90 days in a phase 1 trial, following a single 25 mg dose [36]. In 2022, Vutrisiran received approval in the USA for treating polyneuropathy associated with hereditary transthyretin amyloidosis (ATTRv) in adults.

The ongoing HELIOS-A study (NCT03759379) is a global, randomized, open-label phase 3 trial [37] designed to evaluate the efficacy and safety of vutrisiran compared to an external placebo in ATTRv patients. Participants were randomly assigned in a 3:1 ratio to receive either subcutaneous vutrisiran at a dose of 25 mg every three months or intravenous patisiran at 0.3 mg/kg every three weeks for 18 months. The study included 164 participants, with 122 receiving vutrisiran and 42 receiving patisiran, alongside an external placebo group of 77 individuals. At the nine-month point, Vutrisiran met its primary endpoint by demonstrating a significant change from baseline in the mNIS + 7 score (*p* = 3.54 × 10^−12^) and achieved all secondary efficacy endpoints, including the Norfolk QoL-DN scale, 10-Meter Walk Test, modified BMI, reduction in serum TTR levels, and the Rasch-built Overall Disability Scale. The reduction in TTR levels with vutrisiran was found to be at least as effective as that of patisiran.

The HELIOS-B Trial (NCT04153149) [38] is a Phase 3 randomized, double-blind, placebo-controlled study in which researchers assigned patients with ATTR Cardiomyopathy (either ATTRv or ATTRwt) in a 1:1 ratio to receive either vutrisiran (25 mg) or a placebo every 12 weeks, with treatment lasting up to 36 months. A total of 655 participants were randomized; 326 received vutrisiran while 329 were given the placebo. The primary outcome was a combination of All-Cause Mortality and the occurrence of recurrent cardiovascular events. Treatment with vutrisiran was linked to a lower risk of death from any cause and reduced rates of recurrent cardiovascular events compared to the placebo group. Among the entire population, those receiving vutrisiran experienced a smaller reduction in the distance covered on the 6 min walking test compared to those on placebo (least-squares mean difference of 26.5 m; 95% CI, 13.4 to 39.6; *p* < 0.001), as well as a lesser decline in the KCCQ-OS score (least-squares mean difference of 5.8 points; 95% CI, 2.4 to 9.2; *p* < 0.001) (Table 2).

Inotersen is a type of ASO administered via subcutaneous injection that attaches to human TTR RNA, leading to a reduction in the production of TTR by the liver. Research conducted on a mouse model with the human *TTR* Ile84Ser mutation revealed that *TTR* ASOs reduced hepatic TTR mRNA and serum TTR levels by 80% [39]. The Neuro-TTR trial (NCT01737398) was a phase 2/3 randomized, double-blind, placebo-controlled study that involved administering either inotersen or a placebo for 65 weeks to patients suffering from TTR-Mediated Polyneuropathy [40]. The primary outcome measures included changes in the mNIS + 7 score and the Norfolk Quality of Life Diabetic Neuropathy (Norfolk QoL-DN) Questionnaire from baseline. Results showed that the treatment group performed significantly better than the control group in both the mNIS + 7 and quality of life assessments. The efficacy and safety of inotersen were further examined in a two-year open-label study, which indicated a slowing of disease progression and an improvement in quality of life [41]. Like patisiran, inotersen is effective in managing hereditary ATTR cardiomyopathy. Inotersen received approval in the United States in 2018 and is indicated for the treatment of polyneuropathy associated with ATTRv in adult patients.

NEURO-TTRansform (NCT04136184) was a phase 3, open-label, randomized study [42] carried out at 40 sites across 15 countries between December 2019 and April 2023. The study involved 168 adult participants diagnosed with Coutinho stage 1 or 2 ATTRv polyneuropathy and possessing a confirmed *TTR* variant. Participants received subcutaneous eplontersen (a ligand-conjugated antisense medication) at a dose of 45 mg every four weeks (n = 144), while a smaller control group received subcutaneous inotersen (300 mg weekly; n = 24). The primary outcomes included changes in mNIS + 7 scores, alterations in the Norfolk QoL-DN score, and variations in serum TTR levels. At week 65, the eplontersen group showed results indicative of a substantial reduction in serum TTR levels (the adjusted mean percentage decline in serum transthyretin was −81.7% for the eplontersen group compared to −11.2% for the control group, leading to a difference of −70.4% [95% CI, −75.2% to −65.7%; *p* < 0.001]). They also exhibited less neuropathy-related impairment (mNIS + 7 composite score: 0.3 vs. 25.1; difference, −24.8 [95% CI, −31.0 to −18.6; *p* < 0.001]) and an improved quality of life in comparison to a historical placebo (Norfolk QoL-DN score: −5.5 vs. 14.2; difference, −19.7 [95% CI, −25.6 to −13.8]; *p* < 0.001).

CARDIO-TTRansform (NCT04136171) is a current international, double-blind, randomized, placebo-controlled Phase 3 study focusing on cardiovascular outcomes. It aims to assess the effectiveness and safety of eplontersen in over 1400 individuals diagnosed with ATTR cardiomyopathy. Participants will be assigned to receive either eplontersen or a placebo. The primary outcome measure is a combination of cardiovascular deaths and repeated cardiovascular clinical events. Results are expected to be published in 2025 [43].

An alternative approach to mRNA gene silencing has been preclinically evaluated using the CRISPR-Cas9 system to achieve in vivo knockout of the *TTR* gene. A currently ongoing phase 1 trial (NCT04601051) is assessing the safety, tolerability, and pharmacodynamics of NTLA-2001, which consists of a lipid nanoparticle that encapsulates messenger RNA for the Cas9 protein and a single guide RNA directed at *TTR*. This trial involves patients with ATTRv who have polyneuropathy and those with ATTRv who have cardiomyopathy. The primary objectives of the study include: the number of participants experiencing treatment-related adverse events, the number of participants with clinically significant findings from laboratory tests, the number of participants with clinically significant safety metrics, and the percentage change from baseline in serum TTR and prealbumin levels. Initial findings from the study indicate that by day 28, serum TTR levels in patients with ATTRv and polyneuropathy decreased by 52% (ranging from 47% to 56%) in the group that received a 0.1 mg per kilogram dose, while those receiving a 0.3 mg per kilogram dose showed an 87% reduction (ranging from 80% to 96%). Adverse events were generally mild; however, long-term data are still pending [44] (Table 3).

#### 3.2.3. Danon Disease

Danon disease is an X-linked dominant lysosomal and glycogen storage disorder caused by a mutation in a gene called *LAMP2* (Lysosomal-Associated Membrane Protein 2), which encodes a protein that is an essential component of the lysosomial membrane and seems to have a role in the autofagosome-lysosome fusion [45]. The treatment has always been mainly symptomatic, and in some patients, heart transplant is needed, even though it is not a definite cure. In 2019, investigators initiated a non-randomized, open-label phase 1 trial (NCT03882437) to assess the safety of gene therapy utilizing RP-A501, a recombinant AAV9 that includes the *LAMP2B* transgene, in male individuals diagnosed with Danon Disease (see Table 4) [46]. The study, started 7 April 2019, will probably be completed in 2025. Approximately 7–10 male subjects aged 8 and over received a single IV infusion of RP-A501. The primary endpoints are: treatment-related adverse events, cardiomyocyte histologic correction, clinical stabilization of cardiomyopathy assessed with cardiopulmonary testing.

#### 3.2.4. Fabry Disease

Fabry disease (FD) is an X-linked lysosomal storage disorder resulting from pathogenic mutations in the galactosidase alpha (*GLA*) gene [47]. These mutations cause varying degrees of deficiency in *GLA* enzyme activity, leading to the accumulation of glycosphingolipids, such as globotriaosylceramide (Gb3), its deacylated form, and globotriaosylsphingosine (lyso-Gb3), in the heart, kidney, and blood vessels. Current treatment options include intravenous enzyme replacement therapy or oral chaperone therapy, which can alleviate neuropathic pain, gastrointestinal symptoms, and renal dysfunction but are less effective for cardiomyopathy associated with the disease [48]. FD is considered a promising candidate for gene replacement therapy due to its monogenic nature and the potential for clinical benefits with even a modest increase in *GLA* enzyme activity, reaching as low as 10% of normal levels. Several ongoing studies are assessing the safety and effectiveness of gene therapy approaches for FD.

The INGLAXA open-label Phase 1/2 clinical trial (NCT04519749) is currently assessing the safety and tolerability of gene replacement therapy with 4D-310, an engineered AAV vector that contains the *GLA* transgene and has a strong affinity for cardiomyocytes, in patients affected by classic or late-onset FD with cardiac involvement (Table 5). The main endpoint of the study is to assess the frequency and intensity of adverse events. The secondary endpoints include measuring changes in serum *GLA* activity and serum Gb3 levels. The 4D-310 was effective in raising blood *GLA* levels that can be utilized by adjacent cells, including cardiomyocytes and renal cells [49,50]. Patients receiving 4D-310 demonstrated enhancements in global longitudinal strain and peak VO2 from baseline to the 52-week mark. In situ hybridization revealed that approximately 50% of cardiomyocytes displayed positive transgene expression. Immunohistochemical analysis verified the presence of alpha-galactosidase protein in all samples. However, some patients experienced a temporary episode of acute hemolytic uremic syndrome between days 3 and 7. Following this aHUS-related dose-limiting toxicity, the FDA put the program on clinical hold.

The STAAR trial (NCT04046224) investigates ST-920 (isaralgagene civaparvovec), which is a liver-targeted recombinant AAV vector containing the cDNA for human *GLA*, administered in a single infusion (see Table 5) [51]. This study is ongoing and is classified as a phase 1/2, multicenter, open-label investigation focusing on a single-dose, dose-ranging approach. The main objective is to monitor adverse events related to the treatment. Nine male participants with classic FD and mild to moderate severity were included in the dose escalation phase. No adverse events related to the treatment were reported, and the patients exhibited sustained *GLA* activity. During a follow-up period of up to 25 months, particularly at the highest dosage, a consistent and dose-dependent reduction in lyso-Gb3 levels was observed.

In conclusion, NCT03454893 was an open-label clinical study aimed at assessing the autologous transplantation of CD34+ stem cells that were modified with a lentiviral vector carrying the human *GLA* gene in 15 patients diagnosed with Fabry disease (FD). Regrettably, the enrollment for the study has been stopped following unsatisfactory engraftment results in five of the participants (see Table 5) [52].

#### 3.2.5. Pompe Disease

Pompe disease (PD) is a rare genetic disorder caused by mutations in the gene of the enzyme of acid alpha-glucosidase (*GAA*) responsible for the breakdown of glycogen [53]. In the literature we know about 150 different mutations of *GAA* that cause a lack of function of the enzyme [54]. Five clinical trials have been designed, and the goal of all of them is to provide a functional copy of *GAA* with the use of adenoviruses as vectors (Table 6).

The first clinical trial started in 2010, and it has completed Phase II. This (NCT00976352) was a phase 1/2 trial in which nine ventilator-dependent children were recruited. They received intradiaphragmatic administration of an AAV-mediated *GAA* gene (rAAV1-CMV-hGAA); the primary endpoints were the safety assessments of the rAAV1-CMV-*GAA* and the changes in AAV levels and in *GAA* levels. The patients showed good improvements in the ventilatory performance after the first year of follow up. No one experienced major adverse events related to the study agent; however, six patients experienced capnothorax or pneumothorax related to the technique and the site of the injection [55].

The same team of researchers created an AAV9 vector containing a transgene that encodes human GAA, regulated by a human desmin (*DES*) promoter (rAAV9-DES-hGAA), which was tested in a double-blind, randomized, phase I controlled study (NCT02240407) in patients with late-onset PD. The primary measure tested was the safety of rAAV9-DES-hGAA, while the secondary measures were neurophysiological tests, muscular function, and muscle MRI, which evaluated the effects of injecting this therapy into a leg muscle [56]. The study started in 2017 and ended in 2021, and two patients were studied, but to date there have been no reports on the efficacy and safety data.

A Phase 1/2 prospective, open-label clinical trial (NCT03533673) has used AAV2/AAV8 as vectors of the healthy *GAA* gene (ACTUS-101) to cells in the liver (where functional *GAA* can be built) in patients with late-onset Pompe disease (PD that manifests after the age of one and impacts skeletal muscle, leading to gradual muscle weakness and breathing difficulties). Seven patients with late-onset Pompe Disease have been enrolled till now, and the primary endpoints of the study are the incidence of treatment-related adverse events and the number of patients with abnormal laboratory values. The trial started in 2018, and it should end in 2026 [57].

A Phase 1/2 open-label, ascending dose, multicenter clinical trial (NCT04174105) is testing AT845, an AAV8 vector that is able to transport *GAA* in skeletal muscle and heart, in patients with late-onset Pompe disease. The estimated enrollment number of participants should be 18 patients. The primary measures are safety and tolerability, *GAA* activity, and *GAA* expression. The trial is ongoing, and it should end in 2029 [58].

The RESOLUTE Phase 1/2 dose-escalation trial (NCT04093349) is evaluating the safety and effectiveness of the single infusion of SPK-3006, a liver-tropic AAV containing *GAA* transgene, in adults with clinically moderate late-onset Pompe Disease who have been on enzyme replacement therapy. The primary endpoints are the number of adverse events and the immune response against the AAV and *GAA* transgene. Results ought to be published in 2032.

Gene therapy may serve as a viable and efficient approach for treating Pompe disease. Nonetheless, various aspects of in vivo AAV gene therapy, including dosage, distribution within the body, and especially the immune response to both the vector and the transgene product, require further investigation.

#### 3.2.6. Friedreich’s Ataxia

Friedreich’s Ataxia (FA) is the most common form of human ataxia and arises from insufficient production of the frataxin (*FXN*) protein, typically caused by a triplet expansion in the nuclear *FXN* gene [59]. This gene fails to be transcribed, preventing the formation of the messenger RNA necessary for frataxin synthesis. Frataxin functions as an iron-binding protein located in the mitochondrial matrix [60]. In its absence, various iron-sulfur-dependent proteins within mitochondria and the cytosol do not assemble correctly, leading to impaired mitochondrial and nuclear functions. While the most noticeable clinical symptom is progressive and debilitating ataxia, HCM leading to heart failure is the primary cause of early mortality associated with this condition. Currently, there is no available cure.

There is only one ongoing clinical trial (NCT05302271) evaluating gene therapy for the treatment of cardiomyopathy linked to FA: a phase 1, open-label, dose escalation study of AAVrh.10hFXN, a gene transfer vector using the rh. 10 serotype of an AAV that encodes for *FXN* (Table 7). The aim of this research is to evaluate the safety and initial effectiveness of AAVrh.10hFXN for the treatment of cardiomyopathy linked to Friedreich’s ataxia (FA) in 25 estimated patients. The treatment is delivered through intravenous infusion. The study started in February 2022, and it will end approximately in 2029. Alternative outcome measures include alterations in cardiopulmonary exercise testing, variations in cardiac-related metrics observed in cardiac magnetic resonance imaging and echocardiograms, as well as changes in arrhythmias monitored over a 24 h period.

AAVrh.10hFXN has been assessed earlier in FXN knockout mice [61]. Administering a dose of 1.8 × 10^12^ gc/kg via intravenous injection led to elevated cardiac FXN levels and a notable enhancement in ejection fraction and fractional shortening (*p* < 0.05 for both measures), along with a 21.5% decrease in mortality (*p* < 0.001).

#### 3.2.7. Dilated Cardiomiopathy

Dilated cardiomyopathy (DCM) is marked by left or biventricular dilation and systolic dysfunction that cannot be accounted for by abnormal loading conditions or coronary artery disease [62]. Its etiology may be genetic, estimated to be between 35% and 50%. DCM can be inherited in an autosomal dominant, recessive, or X-linked fashion.

Most cases of isolated DCM follow an autosomal dominant inheritance pattern with age-dependent penetrance and variable clinical expression, which may not manifest until the fifth or sixth decade of life. This suggests the involvement of potential genetic, epigenetic, and environmental modifiers. To date, over 50 disease-related genes have been identified. Despite the vast genetic heterogeneity, Titin (*TTN*) is the most common disease-associated gene, accounting for 20–25% of genetic causes [63]. The second most prevalent gene is lamin A/C (*LMNA*) at around 6%, followed by the beta-myosin heavy chain. Several other genes represent 1–2% of familial cases, while many others are either rare or have been reported only once. Mutations of sarcomeric proteins are also associated with autosomal dominant DCM, even though rare, including actin alpha cardiac muscle 1 (*ACTC1*), *MYBPC3*, *MYH6*, *MYH7* (the most common one), *TNNC1*, *TNNI3*, *TNNT2*, and *TPM1*. Mutations tied to DCM also involve nuclear membrane proteins, such as lamin A/C (*LMNA*) and emerine (*EMD*), an inner nuclear membrane protein associated with Emery-Dreifuss syndrome. Genes regulating transcription play an important role in the DCM, with rare mutations reported in T-box transcription factor 20 (*TBX20*), NK2 homeobox 5 (*NKX2-5*), GATA Binding Protein 4 (*GATA4*), *GATA6*, Forkhead Box D4 (*FOXD4*), PR domain containing 16 (*PRDM16*), EYA transcriptional coactivator and phosphatase 4 (*EYA4*), GATA Zinc Finger Domain Containing 1 (*GATAD1*), and RNA binding motif protein 20 (*RBM20*) [64,65].

Another significant gene is the one encoding for the alpha subunit of the cardiac sodium channel type V (*SCN5A*), which is vital for cardiac action potentials. This channel mediates the rapid depolarization of the myocardium and maintenance of impulse conduction. Mutations in phospholamban (*PLN*) are also present. This protein regulates the process of calcium handling by interacting with the SERCA2 protein [63].

The Z-disc of the sarcomere consists of actin filaments from adjacent sarcomeres linked transversely by α-actinin molecules. Mutations in various Z-disc-associated proteins have been reported in connection with DCM, including α-actinin 2 (*ACTN2*), Cardiac ankyrin repeat protein (*ANKRD1*), Bcl2-associated athanogene 3 (*BAG3*), αB-Crystallin (*CRYAB*), LIM domain-binding protein 3 (*LDB3*), Myopalladin (*MYPN*), Nebulette (*NEBL*), Nexilin (*NEXN*), Telethonin (*TCAP*), Filamin C (*FLNC*), and Desmin (*DES*). Mutations in *DES*, *CRYAB*, *FLNC*, *LDB3*, and *BAG3* lead to myofibrillar myopathies and/or DCM [63]. The potential of gene replacement therapy in treating DCM has been explored exclusively through preclinical studies, with ASOs being a prominent strategy. ASOs are specifically designed to target *PLN* mRNA, downregulating *PLN* activity and its interaction with SERCA2a in heart failure murine models, preventing left ventricular dilatation, improving left ventricular contractility, and leading to an increase in survival rate [66].

An in vitro study assessed the effectiveness of an ASO directed at *PLN*, reducing its impact on SERCA2a in heart cells obtained from nine individuals suffering from advanced heart failure. This study demonstrated an increase in the contractile capacity of these cells, highlighting the potential of this approach to enhance cardiac function by specifically modulating *PLN* levels [67].

Recently Nishiyama et al. [68] have conducted a study on potential genic treatments for dilative cardiomyopathy using types of genomic editing such as base editing (BE) and prime editing (PE). The precise editing of *RBM20* mutation (c.1901 G>A) using adenine base editing through Cas 9 technology brought about an improvement in cardiac function of mice treated.

Another gene therapy strategy evaluated is the spliceosome-mediated RNA trans-splicing as a treatment method for *LMNA*-related congenital muscular dystrophy. The technique involves inserting a corrective RNA sequence that splices with the mutated mRNA, thereby restoring the production of functional *LMNA* proteins. The results demonstrate that this approach can significantly enhance the synthesis of the normal protein and mitigate clinical symptoms in preclinical models [69], and additional studies are required to develop this approach in vivo for DCM.

#### 3.2.8. Duchenne Disease

Duchenne muscular dystrophy (DMD) arises from alterations or losses in the extensive dystrophin gene, which plays a vital role in maintaining the stability of muscle fiber cell membranes. This disorder is passed down through an X-linked recessive inheritance pattern [70]. A promising approach for addressing DMD in patients is delandistrogene moxeparvovec, an AAV vector that delivers a gene encoding a micro-dystrophin protein to the muscles involved in the condition. In the United States, delandistrogene moxeparvovec has received approval for the treatment of ambulatory children aged 4 to 5 years with Duchenne muscular dystrophy (DMD) who have a verified mutation in the DMD gene [71]. The long-term expression of microdystrophin protein from delandistrogene moxeparvovec, which is a truncated form of dystrophin retaining crucial functional domains of the wild-type protein, has shown potential to positively impact in the progression of DMD.

An open-label, phase 1/2 trial (NCT03375164) enrolled four males aged 4 to less than 8 years old with DMD. Patients received one intravenous injection of delandistrogene moxeparvovec along with prednisone. The main endpoint was the number of participants with adverse events. Secondary endpoints were Delandistrogene Moxeparvovec dystrophin expression and the change in the 100-meter timed test. The trial reported 18 treatment-related adverse events, all resolved within 70 days post-treatment. Functional enhancements were maintained over a period of four years, indicating that delandistrogene moxeparvovec might positively influence the progression of the disease (Table 8) [72].

A multicenter, randomized, double-blind, placebo-controlled study, conducted as a two-part crossover trial spanning 48 weeks per segment (NCT03769116), assessed the efficacy of delandistrogene moxeparvovec in patients aged ≥4 to <8 years with DMD (Table 8). Patients were randomly assigned and divided by age to receive either placebo (n = 21) or delandistrogene moxeparvovec (n = 20), and subsequently crossed over for Part 2. The two main endpoints were dystrophin expression and change in North Star Ambulatory Assessment (NSAA) score, which rates the performance of various motor abilities in ambulant children with Duchenne Muscular Dystrophy.

The study demonstrated successful dystrophin expression in all patients (average improvement from baseline to Week 12 of 23.82% in Part 1 and 39.64% in Part 2) and overall stabilization in NSAA score for up to 2 years post-treatment [73].

The ENDEAVOR trial (NCT04626674) is a non-blinded study designed to assess the expression of micro-dystrophin from delandistrogene moxeparvovec, along with its safety and functional results following the delivery of commercially processed delandistrogene moxeparvovec. (Table 8). Eligible ambulatory males aged ≥4 to <8 years received a single intravenous infusion of delandistrogene moxeparvovec. To this day 55 participants have been enrolled. At one year post-treatment, delandistrogene moxeparvovec demonstrated favorable tolerability and showcased stabilized or enhanced motor function, implying potential clinical benefits for patients with Duchenne muscular dystrophy [74].

A phase 1 multicenter (NCT03362502), open-label, non-randomized, single-ascending dose study aims to analyze the tolerability and safety of PF-06939926 (fordadistrogene movaparvovec) gene therapy in ambulatory and non-ambulatory subjects with DMD (Table 8). Fordadistrogene movaparvovec is a recombinant AAV9 containing a truncated human dystrophin gene (mini-dystrophin) controlled by a muscle-specific promoter. The primary measures are treatment-related adverse events, laboratory test abnormalities, change in left ventricular ejection fraction at MRI, and the number of participants with positive responses on the Columbia-Suicide Severity Rating Scale (C-SSRS). The study is currently ongoing, with approximately 22 subjects enrolled to this day [75].

In addition, a phase 2 multicenter single-arm trial (NCT05429372) aims to evaluate the safety and dystrophin expression following the administration of PF-06939926 in male participants with early-stage DMD (Table 8). The main endpoints are: treatment-related adverse events, the number of abnormal laboratory findings, changes in neurological tests, changes in body weight, changes in troponin I, modifications in ECG and echocardiogram findings. This study has currently enrolled around 10 participants aged between 2 and 4 years. There is no placebo arm in this study; the study will probably be completed in 2029 [76].

A phase 3, multi-center, randomized, double-blind, placebo-controlled trial is currently assessing the safety and effectiveness of PF 06939926 for treating DMD (NCT04281485) (see Table 8). To date, 122 patients have been enrolled, with the study expected to conclude around 2029. The trial includes boys aged between 4 and 8 years old. Approximately two-thirds of the participants were placed in Cohort 1 and received gene therapy at the start of the trial, while about one-third belong to Cohort 2, initially receiving a placebo but scheduled to receive gene therapy after one year, contingent upon ongoing safety assessments. The primary outcome measure is the change in the North Star Ambulatory Assessment (NSAA) score. Secondary outcome measures include changes in mini-dystrophin expression, serum creatinine kinase levels, and performance on the 10 m walk test. All participants will be monitored for five years following the administration of gene therapy [77].

The last study is a controlled, open-label, single-ascending dose study (NCT03368742) evaluating the efficacy of SGT-001, a novel AAV9 vector-mediated gene transfer therapy (Table 8). The AAV9 vector contains a muscle-specific promoter microdystrophin developed to deliver and express a shorter but functional form of dystrophin. In this study, a total of 12 participants will undergo one intravenous (IV) administration of SGT-001 and will be monitored for around 5 years. The primary endpoints are changes in microdystrophin protein levels, adverse effects, laboratory and physical abnormalities, and EKG pathological modifications [78].

#### 3.2.9. Arrhythmogenic Cardiomyopathy

Arrhythmogenic cardiomyopathy (ACM) is a cardiomyopathy characterized by fibro-adipous replacement of the myocardium in both the right and left ventricles. The prevalence of this disease ranges from 1 in 2000 to 1 in 5000. Autosomal dominant inheritance is the most common pattern for non-syndromic forms of the disease [79]. Currently, approximately 40–60% of genetic causes are recognized, with plakophilin 2 (*PKP2*) being the most prevalent disease-associated gene [80]. A total of 50% of patients with ACM carry mutations in five different genes that code for desmosomal proteins [81]: Plakoglobin (*JUP*), Desmoplakin (*DSP*), *PKP2*, Desmocollin (*DSC2*), and Desmoglein (*DSG2*).

Desmosomes are cell junctions that are linked to the intermediate filament system, which is predominantly formed by desmin in the heart, and desmin can also be mutated in ACM. Mutations associated with ACM have also been identified in αT-Catenin (*CTNNA3*) and N-cadherin (*CDH2*), which encode for adherens junction proteins [63].

In the last decade, some pre-clinical trials have been completed. In particular, *PKP2* mutant mice have been tested with LX2020, an AAV vector encoding the *PKP2* gene and designed to intravenously deliver a fully functional *PKP2* gene to cardiac muscle. Mice achieved great improvements of left and right ventricle ejection fraction and reduction in right and left end-diastolic volumes. The study improved survival as well [82,83]. Because of the good results of the pre-clinical trials, three clinical trials started.

A Phase 1/2, open-label, dose-escalating, multicenter trial (NCT06109181) is evaluating the efficacy to suppress ventricular arrhythmias and the safety of LX2020 in patients with ACM and a mutation in the *PKP2* gene. The clinical trial started in February 2024; ten patients are expected to be enrolled, and one year will be the duration of follow-up. The primary endpoint is the percentage of patients with at least one treatment-adverse event. Results are expected by 2027 (Table 9) [84].

In August 2023, Phase 1 dose-escalation clinical trial (NCT05885412) of a single-dose intravenous infusion of RP-A601, a recombinant viral vector composed of an adenovirus capsid encapsulating the transgene of the human *PKP2*, in adults affected by high-risk ACM and with a pathogenic *PKP2* truncating variant, started. The study will evaluate the efficacy and the safety of nine patients during the 12 months after the infusion. Results are expected by 2026 [85].

Another clinical trial (RIDGE-1) (NCT06228924) started in March 2024, and it is evaluating the safety and efficacy of TN-401 in patients with symptomatic *PKP2*-ACM. TN-401 is a Recombinant AAV9 containing the *PKP2* transgene. Fifteen patients are expected to be enrolled in two cohorts, and as the previous ones, follow up will last one year. The dose of TN-401 for Cohort 1 will be 3 × 10^13^ vg/kg, and the dose for Cohort 2 will be 6 × 10^13^ vg/kg [86].

At the same time, the RIDGE-1 researchers have designed an observational trial to evaluate the presence of pre-existing antibodies to AAV9 in a population of patients with *PKP2* gene-associated ACM (NCT06311708). The trial should enroll about 200 patients with a confirmed mutation in the *PKP2* gene and a diagnosis of ACM, and clinical observation with annual sample collections will last five years [87]. The aim of the trial is to predict the incidence of autoimmune response that could be a major obstacle to a successful study. Furthermore, we will know in a few years the results of the clinical studies mentioned above, and we expect many others are close to starting.

## 4. Ethical and Safety Considerations

Patients receiving gene therapy need to have a comprehensive understanding of the procedure, encompassing its risks, advantages, and uncertainties. The intricate nature of gene-editing technologies may hinder patients’ ability to give truly informed consent.

From an ethical point of view, there is a concern that gene therapies may become accessible only to those with financial means, thereby worsening current health inequalities. Ensuring equitable access to these treatments is a critical ethical responsibility [88].

The enduring safety of gene therapy is still questionable, as the potential for unforeseen complications or side effects raises ethical issues regarding the accountability of researchers and healthcare providers in safeguarding patient welfare. The lasting impacts of gene therapies, particularly those related to germline editing, remain uncertain. Ethical frameworks must consider how these treatments could affect future generations [89].

Furthermore, a major safety concern in gene therapy are off-target effects, where gene-editing tools unintentionally alter unintended sections of the genome. This can result in harmful mutations, potentially leading to cancers or other health issues. Insertional mutagenesis (when a new gene is inserted into the genome, disrupting existing genes or regulatory elements) may result in cancers or impair the function of essential genes [89].

Gene therapy may trigger an immune response against the vector (often a viral agent) used for delivering the therapeutic gene, causing inflammation or other negative effects. Such immune responses can reduce the therapy’s effectiveness or pose risks to patient health [90].

To ensure that gene therapy research and applications are carried out ethically, strong regulatory frameworks are needed. Oversight should strike a balance between fostering innovation and ensuring patient safety, while also addressing public apprehensions regarding the appropriate use of these technologies.

## 5. Future Perspectives

Innovative technologies such as base editing show great potential for overcoming the challenges faced by existing gene therapy techniques in cardiomyopathies. Traditional gene therapies typically utilize viral vectors to transport genetic material. This approach can result in inefficiencies, immune reactions, and difficulties in precisely reaching targeted tissues, particularly in the heart. Standard gene therapies might struggle to effectively address specific mutations associated with cardiomyopathies, particularly point mutations that affect single nucleotides [91].

Base editing presents several benefits that may help mitigate these issues. It permits the conversion of one base pair to another without inducing double-strand breaks. This level of precision reduces the likelihood of off-target effects, thereby enhancing the safety of genetic alterations. For cardiomyopathies, this could enable the correction of specific mutations that cause the condition [92].

Innovations in nanoparticle-based delivery technologies and modified viral vectors could boost targeting precision and efficacy in cardiac tissues. When paired with base editing methods, these systems may enable more accurate delivery to cardiac myocytes, improving treatment outcomes [93].

Future developments in base editing could permit adjustable and reversible genetic modifications. This flexibility could be advantageous for therapies addressing conditions like certain progressive forms of cardiomyopathies that may require periodic reassessments.

Integrating base editing with other therapeutic approaches, such as small molecules or RNA-based techniques, could boost overall treatment effectiveness [94].

While promising, base editing still faces several hurdles. The potential long-term impacts of base editing require thorough research; gaining public trust in gene editing technologies and creating standardized processes is crucial for making these therapies accessible to a wider patient population.

## 6. Conclusions

Gene therapy is constantly evolving and presents a highly promising therapeutic alternative in the field of cardiomyopathies. Some gene therapies are already available on the market, while others still need to be developed. With ongoing research, validation, and development, these advancements may lead to improved treatment effectiveness, customized therapies for individual patients, and ultimately enhance the quality of life for those affected by cardiomyopathies. Continued collaboration among researchers, clinicians, patients, and regulatory agencies will be vital in realizing these opportunities.

## Figures and Tables

**Figure 1 ijms-25-13147-f001:**
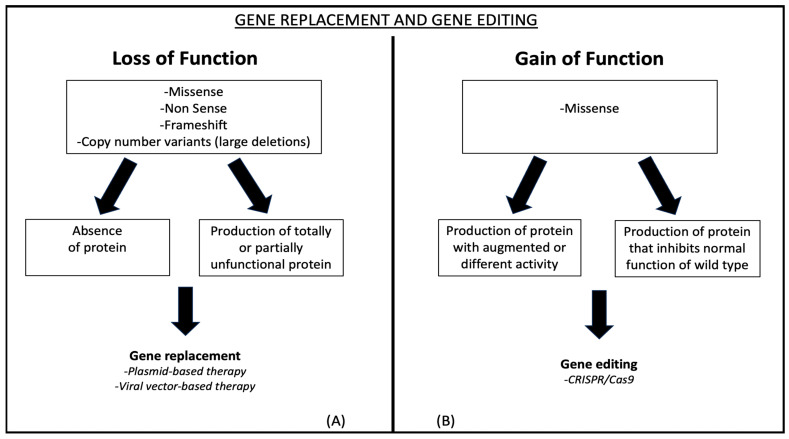
Exploring types of Gene Therapies. (**A**) illustrates how nonsense, frameshift, and missense mutations can lead to loss-of-function mutations, resulting in the absence of functional proteins. (**B**) depicts how missense variants often lead to gain-of-function mutations, resulting in proteins that exhibit altered or enhanced activity compared to their wild-type forms.

**Table 1 ijms-25-13147-t001:** Overview of various clinical trials about Hypertrophic Cardiomyopathy, providing descriptions of their objectives and key findings.

Study Identifier	Description	Status
NCT05836259 (non-randomized open-label study)	Evaluates the safety, tolerability, and pharmacodynamics of TN-201 (Recombinant Adeno-associated Virus Serotype 9 (AAV9) containing Myosin Binding Protein C Transgene) in adult patients (6–30) with symptomatic *MYBPC3* mutation-associated nHCM.	Ongoing

**Table 2 ijms-25-13147-t002:** Overview of various clinical trials about Cardiac Amyloidosis providing descriptions of their objectives and key findings.

Study Identifier	Description	Status
NCT04601051 (Non-Randomized open- label Phase 1 Study)	NTLA-2001 is injected once in six patients with ATTRv with polyneuropathy and patients with ATTRv with cardiomyopathy.	Ongoing
NCT01960348 (APOLLO-A) (Phase 3 double-blind placebo-controlled Study)	ATTRv patients with polyneuropathy were assigned in a 2:1 ratio to receive intravenous patisiran or placebo once every 3 weeks. A total of 225 patients underwent randomization. The patisiran-treated patients had an important improvement in the neuropathy-related measurements and in quality of life.	Terminated
NCT03997383 (APOLLO-B) (Phase 3, Randomized, Double-blind trial)	Patients with ATTRv or ATTRwt were assigned in a 1:1 ratio to receive either patisiran or a placebo, administered every three weeks for a duration of 12 months. A total of 360 individuals were randomly allocated to receive either patisiran or a placebo. After 12 months, the decrease in the distance covered during the 6-MWT was not as significant in the patisiran group as it was in the placebo group. Additionally, the KCCQ-OS scores showed improvement in the patisiran group, whereas they decreased in the placebo group.	Ongoing
NCT03759379 (HELIOS-A) (Phase 3 Global, Randomized, Open-label Study)	ATTRv patients were assigned randomly in a 3:1 ratio to receive subcutaneous vutrisiran every three months or intravenous patisiran every three weeks over a period of 18 months. The study included 164 individuals. Vutrisiran successfully achieved the main objective of altering the baseline score in the mNIS + 7; notable enhancements compared to an external placebo were noted in the Norfolk QoL-DN assessment, the 10 m walking test, modified BMI, and the Rasch-built Overall Disability Scale.	Ongoing
NCT04153149 (HELIOS-B) (Phase 3 randomized, double-blind, placebo-controlled study)	Patients diagnosed with ATTR-CM were administered either vutrisiran or a placebo every 12 weeks in a 1:1 ratio, with the treatment lasting up to 36 months. A total of 655 individuals were randomly assigned; 326 were treated with vutrisiran, while 329 received the placebo. The use of vutrisiran was linked to a decreased likelihood of death from any reason and lower occurrences of repeated cardiovascular events compared to the placebo group, along with a reduced risk of mortality observed up to 42 months.	Ongoing

**Table 3 ijms-25-13147-t003:** Overview of various clinical trials about Cardiac Amyloidosis providing descriptions of their objectives and key findings.

Study Identifier	Description	Status
NCT01737398 (NEURO-TTR) (phase 2/3 randomized, double-blind, placebo-controlled study)	Individuals diagnosed with stage 1/2 ATTRv with polyneuropathy were randomly divided in a 2:1 ratio to receive either weekly subcutaneous injections of inotersen or a placebo. In total, 172 participants (112 in the inotersen group and 60 in the placebo group) received at least one dose of the treatment being studied. Both mNIS + 7 and the Norfolk QOL-DN score showed a preference for inotersen.	Terminated
NCT04136184 (NEURO-TTRansform) (Phase 3 Global, Open-Label, Randomized Study)	Patients with ATTRv with polyneuropathy were randomly assigned to receive subcutaneous injections of either eplontersen every four weeks or inotersen weekly. The group receiving eplontersen showed results indicative of a substantial reduction in serum transthyretin levels, reduced neuropathic impairment, and an improved quality of life when compared to historical placebo data.	Terminated
NCT04136171 (CARDIO-TTRansform) (Phase 3 Global, Double-Blind, Randomized, Placebo-Controlled Study)	Assesses the effectiveness and safety of 1400 individuals diagnosed with ATTR cardiomyopathy, who will be randomly assigned to receive subcutaneous injections of either eplontersen or a placebo once every four weeks.	Ongoing

**Table 4 ijms-25-13147-t004:** Overview of a clinical trial on Danon Disease.

Study Identifier	Description	Status
NCT03882437 (non-randomized open-label phase 1)	A total of 7–10 male subjects aged 8 and over received a single IV infusion of RP-A501 (*LAMP2B* transgene contained in a recombinant adeno-associated virus serotype 9 (AAV9).	Ongoing

**Table 5 ijms-25-13147-t005:** Overview of clinical trials on Fabry Disease.

Study Identifier	Description	Status
NCT04519749 (Open-label, phase 1/2)	The study assesses the safety and tolerability of 4D-310 (AAV vector containing the *GLA* transgene) in patients with classic or late-onset FD with cardiac involvement, regardless of whether they are receiving enzyme replacement therapy with corticosteroid prophylactic immunosuppression.	Ongoing
NCT04046224 (Phase 1/2) STAAR	A total of 13 individuals were analyzed. Using ST-920 (isaralgagene civaparvovec), a functional *GLA* gene is introduced in the liver through a single injection of a recombinant AAV2/6 vector, eliminating the necessity for immunosuppressive therapy.	Ongoing
NCT03454893 (Open-label)	Autologous stem cell transplantation involving CD34+ cells that have been modified in patients with Fabry disease using a lentiviral vector that carries the human *GLA* gene.	Terminated

**Table 6 ijms-25-13147-t006:** Overview of various clinical trials about Pompe Disease providing descriptions of their objectives and key findings.

Study Identifier	Description	Status
NCT00976352 (Phase 1/2)	Nine ventilator-dependent children have received intradiaphragmatic administration of AAV-mediated *GAA* gene	Terminated
NCT03533673 (Prospective, open-label phase 1/2)	The study employed AAV 2/8 as a vector for the *GAA* gene, regulated by a liver-specific promoter (AAV8-LSPhGAA), in seven individuals diagnosed with late-onset Pompe Disease.	Ongoing
NCT04174105 (Phase 1/2 open-label, ascending dose, multicenter clinical study) FORTIS	The trial tests AT845 (AAV8 that is able to express *GAA* in skeletal muscle and heart).	Ongoing
NCT04093349 (Phase 1/2 dose-escalation trial) RESOLUTE	The study evaluates the safety, tolerability, and effectiveness of a single intravenous infusion of SPK-3006 (an AAV vector featuring a bioengineered Rh74-derived capsid that promotes the production of *GAA*) in adults diagnosed with clinically moderate, late-onset Pompe disease who are undergoing enzyme replacement therapy (ERT).	Ongoing
NCT02240407 (double-blind, randomized, phase I controlled study)	The study assesses the toxicity, distribution within the body, and possible effects of re-administering rAAV9-DES-hGAA delivered through intramuscular injection. Two individuals diagnosed with Late-Onset Pompe disease participated in the trial. As of now, there have been no findings regarding the effectiveness and safety of the treatment..	Terminated

**Table 7 ijms-25-13147-t007:** Overview of various clinical trials about Friedreich’s Ataxia.

Study Identifier	Description	Status
NCT05302271 (phase 1, open-label, dose escalation)	Assesses the safety and effectiveness of AAVrh.10hFXN (a gene transfer vector based on the rh.10 serotype adeno-associated virus that encodes *FXN*) for addressing the cardiomyopathy linked to Friedreich’s ataxia.	Ongoing

**Table 8 ijms-25-13147-t008:** Overview of various clinical trials about Duchenne Muscular Dystrophy providing descriptions of their objectives and key findings.

Study Identifier	Description	Status
NCT03375164 (Phase 1/2 open-label)	Four boys between the ages of 4 and 8 years old, with DMD were administered a one-time intravenous infusion of Delandistrogene moxeparvovec (SRP-9001) in combination with prednisone.	Terminated
NCT03769116 (Multicenter, randomized, double-blind, placebo-controlled)	Effectiveness of Delandistrogene moxeparvovec in individuals aged over 4 and under 8 years with DMD. Participants were randomly allocated to receive either a placebo (n = 21) or Delandistrogene moxeparvovec (n = 20), and then switched treatments for Part 2 of the study.	Terminated
NCT04626674 (Open-label phase 1) ENDEAVOR	Assesses the expression of micro-dystrophin, safety, and functional results following the administration of commercially manufactured Delandistrogene moxeparvovec. Males with DMD who were between 4 and 8 years old were given a single intravenous dose of Delandistrogene moxeparvovec (1.33 × 10^14^ vg/kg).	Ongoing
NCT03362502 (open-label, non-randomized, single-ascending dose)	Evaluates the safety and tolerability of PF-06939926 (fordadistrogene movaparvovec) gene therapy in 22 subjects. This therapy utilizes an AAV9 containing a truncated human dystrophin gene (mini-dystrophin) controlled by a muscle-specific promoter.	Ongoing
NCT05429372 (Phase 2 Experimental)	Evaluates the safety and dystrophin expression following the administration of PF-06939926 in male participants with early-stage Duchenne muscular dystrophy (aged between 2 and 4 years).	Ongoing
NCT04281485 (Multicenter, randomized, double-blind, placebo-controlled, phase 3)	Assesses the efficacy of PF-06939926 gene therapy. A total of 122 patients (>4 years old; <8 years old) have been randomly assigned to one of two groups: two-thirds of the participants underwent gene therapy at the beginning of the study, while one-third were given a placebo initially and then received gene therapy after a year.	Ongoing
NCT03368742 (Phase 1/2, controlled, open-label, single-ascending dose)	SGT-001 (AAV9 vector containing muscle-specific promoter and microdystrophin construct) will be injected in 12 patients with DMD who will be followed for approximately 5 years. The study will evaluate its safety, tolerability, and efficacy.	Ongoing

**Table 9 ijms-25-13147-t009:** Overview of various clinical trials about Arrhythmogenic Cardiomyopathy providing descriptions of their objectives and key findings.

Study Identifier	Description	Status
NCT05885412 (Phase 1, dose escalation)	The trial evaluates an intravenously injected recombinant AAV vector containing PKP2 (RP-A601) in subjects with high risk *PKP2*-ACM.	Ongoing
NCT06109181 (Phase1/2, open-label, dose-escalating, multicenter trial)	The study assesses the safety and tolerability of LX2020 (AAV vector encoding *PKP2* gene) in 10 adult patients with *PKP2*-ACM.	Ongoing
NCT06228924 (RIDGE-1) (open-label, phase 1)	The trial will include a maximum of 15 patients across two designated dose groups who are experiencing symptomatic *PKP2*-ACM. Patients in each cohort will receive a single i.v. dose of TN-401 (AAV9 containing *PKP2* Transgene).	Ongoing
NCT06311708 (multicenter, observational)	The study evaluates the prevalence of pre-existing antibodies to AAV9 in a population of patients with *PKP2*-ACM	Ongoing

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
