# Peer review of "Unveiling the Future of Cardiac Care: A Review of Gene Therapy in Cardiomyopathies"

_ijms, 2024, doi:10.3390/ijms252313147_

Round 1

Reviewer 1 Report

Comments and Suggestions for Authors

In the review article ‘Unveiling the Future of Cardiac Care: A Review of Gene Therapy in Cardiomyopathies’ submitted by Venturiello et al. to IJMS the authors summarize gene therapy in the context of cardiomyopathies and related syndromic diseases including Duchenne dystrophy.

The topic of this review article is highly relevant to a broad readership. However, I have several points, where the authors should extent their manuscript.

1.)   Introduction: In the introduction I would prefer that the authors summarize in more detail the landscape of genetic cardiomyopathies. The book chapter ‘The Genetic Landscape of Cardiomyopathies’ (Gerull 2019) could be helpful in this context.

2.)   Please re-write some parts of the manuscript to reduce the percent overlap with other publications (see iThenticate report).

3.)   Figure 1: I would remove the examples of the specific mutations, since this is clear to nearly all readers.

4.)   Line 95-97: I would also explain novel developments in the context of AAVs like MyoAAV and AAV. Please see the following references for this point:

·         ‘Directed evolution of a family of AAV capsid variants enabling potent muscle-directed gene delivery across species’ (Tabebordbar et al. 2021).

·         ‘Identification of a myotropic AAV by massively parallel in vivo evaluation of barcoded capsid variants’ (Weinmann et al. 2020).

5.)   In Figure 1, I would also include copy number variants (large deletions). Recently, large deletions were identified for example in DSG2 (see ‘Hemi- and Homozygous Loss-of-Function Mutations in DSG2 (Desmoglein-2) Cause Recessive Arrhythmogenic Cardiomyopathy with an Early Onset’ 2021).

6.)   In general, I would suggest to write all human genes in Italics in the complete manuscript and write protein names in regular writing.

7.)   Line 173-176: I would shortly discuss also base editing. Recently base editing was used to correct mutations in RBM20 for example (see ‘Precise genomic editing of pathogenic mutations in RBM20 rescues dilated cardiomyopathy’ Takahiko Nishiyama et al).

8.)   Line 385-387. Please add here also the DES gene encoding desmin (see ‘Functional characterization of the novel DES mutation p.L136P associated with dilated cardiomyopathy reveals a dominant filament assembly defect’ 2016). It is well known, hat DES mutations cause in about 2% of all cases severe DCM.

9.)   Paragraph 3.2.9: Again, the genetic background of ACM has to be explained in more detail. The review article ‘Insights Into Genetics and Pathophysiology of Arrhythmogenic Cardiomyopathy.’ (2021) could be helpful in this context. In minimum, I would shortly explain the desmosomal genes (PKP2, JUP, DSP, DSG2 and DSC2).

10.) An outlook about novel technical improvements would be helpful. What are future research questions in gene therapy of cardiomyopathies? In addition, I would shortly discuss also limitations of gene therapy approaches.

In summary, I suggest a major revision for this manuscript. The topic is really interesting. However, there are several parts which needs extension, since relevant background is not explained. However, I wish the authors good luck!

Author Response

We are pleased to have the opportunity to revise our paper “Unveiling the Future of Cardiac Care: A Review of Gene Therapy in Cardiomyopathies” for publication in the Journal.

We truly appreciated the detailed and thoughtful criticisms of the reviewer. His comments have been addressed below point-by-point in this rebuttal. The changes in the manuscript are marked in red.

We thank the reviewer for the consideration of our manuscript and for the accurate annotations.

1)  “Introduction: In the introduction I would prefer that the authors summarize in more detail the landscape of genetic cardiomyopathies. The book chapter ‘The Genetic Landscape of Cardiomyopathies’ (Gerull 2019) could be helpful in this context.”

We agree with the comment. We have modified the text by adding informations related to the genetics of each cardiomyopathy, taking inspiration from the chapter ‘The Genetic Landscape of Cardiomyopathies’ by Gerull of the book ‘Genetic Causes of Cardiac Disease’ by J. Erdmann and A. Moretti (recommended by the reviewer). However, the additions have not been included in the introduction (to keep it from becoming too lengthy), but rather in each chapter related to the individual cardiomyopathies.

2) “Please re-write some parts of the manuscript to reduce the percent overlap with other publications (see iThenticate report).”

Sure, we did it.

3)   “Figure 1: I would remove the examples of the specific mutations, since this is clear to nearly all readers.”

 Sure, we did it.

4) “Line 95-97: I would also explain novel developments in the context of AAVs like MyoAAV and AAV. Please see the following references for this point:

  • ‘Directed evolution of a family of AAV capsid variants enabling potent muscle-directed gene delivery across species’ (Tabebordbar et al. 2021).
  • ‘Identification of a myotropic AAV by massively parallel in vivo evaluation of barcoded capsid variants’ (Weinmann et al. 2020).”

 We thank the reviewer for this comment.

We drew inspiration from the cited articles to add a paragraph on MyoAAV at line 99 page 3 of the text.

 5) “In Figure 1, I would also include copy number variants (large deletions). Recently, large deletions were identified for example in DSG2 (see ‘Hemi- and Homozygous Loss-of-Function Mutations in DSG2 (Desmoglein-2) Cause Recessive Arrhythmogenic Cardiomyopathy with an Early Onset’ 2021).”

We thank the reviewer for having raised this point. We added in the figure copy number variants since we agree that is a very important topic in the genetics of cardiomyopathies.

6) “In general, I would suggest to write all human genes in Italics in the complete manuscript and write protein names in regular writing.”

We appreciate this accurate annotation and we wrote all genes in Italics.

7) “Line 173-176: I would shortly discuss also base editing. Recently base editing was used to correct mutations in RBM20 for example (see ‘Precise genomic editing of pathogenic mutations in RBM20 rescues dilated cardiomyopathy’ Takahiko Nishiyama et al).”

 We agree; base editing has become more and more a major in gene therapy. We added a paragraph on base editing and Nishiyama’s study at line 507 page 16 of the text.

8) “Line 385-387. Please add here also the DES gene encoding desmin (see ‘Functional characterization of the novel DES mutation p.L136P associated with dilated cardiomyopathy reveals a dominant filament assembly defect’ 2016). It is well known, hat DES mutations cause in about 2% of all cases severe DCM.”

We thank for the point. We added notions on Desmin at line 495 page 16 talking about dilated cardiomyopathy and at line 608 page 20 talking about ACM.

9) “Paragraph 3.2.9: Again, the genetic background of ACM has to be explained in more detail. The review article ‘Insights Into Genetics and Pathophysiology of Arrhythmogenic Cardiomyopathy.’ (2021) could be helpful in this context. In minimum, I would shortly explain the desmosomal genes (PKP2, JUP, DSP, DSG2 and DSC2).”

 We agree very much with your point. We added many notions on the genetic background of ACM at the beginning of the ACM chapter (line 603 page page 20).

10) “An outlook about novel technical improvements would be helpful. What are future research questions in gene therapy of cardiomyopathies? In addition, I would shortly discuss also limitations of gene therapy approaches.”

 We thank you for this comment. We agree these are very important subjects to address and we added two entire chapters discussing the topics, with their corresponding references (chapter 4 and 5, pages 21-22)

Reviewer 2 Report

Comments and Suggestions for Authors

The manuscript entitled "Unlocking the future of cardiac therapy: a review of gene therapy in cardiomyopathies" provides a comprehensive analysis of current gene therapy strategies for cardiomyopathies. The authors systematically review various therapeutic modalities, including gene replacement, genome editing and RNA interference approaches, with particular emphasis on their applications in both hypertrophic and dilated cardiomyopathies. The review follows a well-structured framework, moving seamlessly from basic concepts to specific clinical applications, recent trials, and future developments in the field. The authors effectively synthesize findings from pivotal clinical and preclinical studies while articulating the transformative potential of gene therapy in cardiac medicine. Their critical analysis of future challenges, particularly with regard to immune responses and delivery optimization, provides valuable insights for the field. While this review makes a substantial contribution to the current literature, several key areas warrant further elaboration.

1. Clarify Clinical Trial Details: Include additional specificities about trial designs (e.g., control groups, endpoints) to facilitate comparison among different trials discussed.

2. Address Ethical and Safety Concerns. Discuss ethical considerations and potential safety concerns associated with gene therapy, especially regarding long-term effects and off-target gene editing risks.

3. Incorporate Quantitative Data for Results: Supplement the results section with quantitative data (e.g., percentages of gene expression changes or survival rates) for a more robust understanding of efficacy outcomes.

4. Provide Future Perspectives on Emerging Techniques: Include a section that projects how emerging technologies, such as base editing and prime editing, could potentially address the limitations of current gene therapy methods in cardiomyopathy.

Author Response

We are pleased to have the opportunity to revise our paper “Unveiling the Future of Cardiac Care: A Review of Gene Therapy in Cardiomyopathies” for publication in the Journal.

We truly appreciated the detailed and thoughtful criticisms of the reviewer. His comments have been addressed below point-by-point in this rebuttal. The changes in the manuscript are marked in red.

We thank the reviewer for the consideration of our manuscript and for the accurate annotations.

1)  “Clarify Clinical Trial Details: Include additional specificities about trial designs (e.g., control groups, endpoints) to facilitate comparison among different trials discussed.”

We thank you very much for your comment. At first we didn’t add additional specificities about trial designs because we didn’t want to make the subject too lengthy, but we think you’re right. It’s important to explain endpoints and the design of each study to better understand the differences between them. We added these informations for each trial in the text.

2) “Address Ethical and Safety Concerns. Discuss ethical considerations and potential safety concerns associated with gene therapy, especially regarding long-term effects and off-target gene editing risks.”

We thank you for this consideration. We agree very much with this point. That’s why we added an entire chapter (chapter 4) on this topic right before the conclusions. You can find it at page 21 of the text.

3)   “ Incorporate Quantitative Data for Results: Supplement the results section with quantitative data (e.g., percentages of gene expression changes or survival rates) for a more robust understanding of efficacy outcomes.”

We thank you for this comment. We added quantitative data available for each clinical trial. Specifically we added the main endpoints for each trial and the changes in the endpoint percentages after the study.

4) “Provide Future Perspectives on Emerging Techniques: Include a section that projects how emerging technologies, such as base editing and prime editing, could potentially address the limitations of current gene therapy methods in cardiomyopathy.”

We thank you for this comment. We agree with this point. We added an entire chapter (chapter 5) on the topic right before the conclusions. You will find it on page 22 of the text.

Round 2

Reviewer 1 Report

Comments and Suggestions for Authors

The authors have improved their revised manuscript according to my suggestions. In summary I suggest to accept this manuscript fir publication.